# High-throughput single-cell chromatin accessibility CRISPR screens enable unbiased identification of regulatory networks in cancer

Sarah E. Pierce[1,4], Jeffrey M. Granja [1,2,4] & William J. Greenleaf [1,2,3✉]

Chromatin accessibility profiling can identify putative regulatory regions genome wide; however, pooled single-cell methods for assessing the effects of regulatory perturbations on accessibility are limited. Here, we report a modified droplet-based single-cell ATAC-seq protocol for perturbing and evaluating dynamic single-cell epigenetic states. This method (Spear-ATAC) enables simultaneous read-out of chromatin accessibility profiles and integrated sgRNA spacer sequences from thousands of individual cells at once. Spear-ATAC profiling of 104,592 cells representing 414 sgRNA knock-down populations reveals the temporal dynamics of epigenetic responses to regulatory perturbations in cancer cells and the associations between transcription factor binding profiles.

[1] Department of Genetics, Stanford University School of Medicine, Stanford, CA, USA. [2] Center for Personal and Dynamic Regulomes, Stanford University School of Medicine, Stanford, CA, USA. [3] Chan Zuckerberg Biohub, San Francisco, CA, USA. [4]These authors contributed equally: Sarah E. Pierce, Jeffrey M. Granja. ✉email: wjg@stanford.edu

Complex epigenetic regulation is a unique requirement for all multicellular organisms, enabling diverse phenotypes stemming from a common underlying genotype. Understanding how transcription factor binding dynamics drive epigenetic state remains one of the durable mysteries of cell biology, underlying fundamental process such as embryogenesis, differentiation, and cancer[1]. Perturbing the expression levels of epigenetic regulators and observing the subsequent effects on chromatin accessibility provides a powerful means to dissect transcription factor function. To this end, CRISPR/Cas9 technologies enable precise tuning of protein levels using targeted mutation and epigenetic modulation strategies[2,3]. CRISPR/Cas9 perturbation methods allow knock-down of protein levels of any gene, and when combined with technologies such as scRNA-seq, the global transcriptional effects of these perturbations can be assayed across thousands of cells for each perturbation[4–6]. In contrast, however, current methods to profile the effects of CRISPR/Cas9-based perturbations on single-cell epigenomes are limited to analyzing 96 cells per run on an integrated fluidic circuit[7].

Here, we introduce the droplet-based Spear-ATAC protocol (Single-cell perturbations with an accessibility read-out using scATAC-seq) to quantify and map the effects of perturbing transcription factor levels on chromatin accessibility in high throughput. In contrast to previous methods, Spear-ATAC relies on reading out sgRNA spacer sequences directly from genomic DNA rather than off of RNA transcripts. We use Spear-ATAC to evaluate epigenetic responses to perturbations across time as well as to understand the effect of transcription factor perturbation on the accessibility of transcription factor binding profiles.

## Results

**Single-cell CRISPR screens with a chromatin accessibility read-out.** Similar to bulk accessibility profiling using ATAC-seq[8,9], the droplet-based scATAC-seq protocol begins with nuclei isolation and transposition of the sample of interest using a hyperactive transposase (Tn5) that integrates into areas of open chromatin[10,11] (Fig. 1a and Supplementary Fig. 1). To first guarantee that capture of a single-copy of an integrated sgRNA does not depend on the local accessibility context surrounding the sgRNA sequence, we flanked the lentiviral sgRNA spacer with pre-integrated Nextera Read1 and Read2 adapters (Supplementary Fig. 2a–b), ensuring that the sgRNA sequence will amplify with the same primers used to amplify ATAC-seq fragments in the library. When testing the detection of sgRNA fragments with bulk ATAC-seq, this design increased our ability to detect sgRNA fragments by ~4-fold without altering sgRNA efficacy (Supplementary Fig. 2c–d). Following transposition, the nuclei are loaded into the 10x Controller for the capture of individual nuclei into nanoliter-scale gel-beads in emulsion (GEMs). These GEMs contain barcoded Forward oligos complementary to the Nextera Read1 adapter to amplify all ATAC fragments, thereby tagging each ATAC fragment from the same nucleus with the same 10x barcode. Since this protocol uses a single barcoded primer to tag ATAC fragments, we reasoned that we could preferentially amplify each sgRNA fragment by also spiking in a Reverse oligo specific to the sgRNA backbone (Fig. 1a and Supplementary Fig. 3a–b). This modification allows for exponential amplification of the sgRNA fragment at the same time the rest of the library is amplified linearly, while still ensuring the sgRNA can be used as a substrate for the second round of PCR. We also extended the number of cycles of in-GEM linear amplification of scATAC-seq fragments from 12 to 15, which subsequently adds three rounds of exponential sgRNA amplification without altering scATAC-seq

quality (Supplementary Fig. 3c–d). Finally, we included a biotin-tagged primer during targeted sgRNA amplification, which allows for the specific enrichment of these fragments while minimizing aberrant background signal from scATAC-seq reads (Supplementary Fig. 3e). Overall, these changes increase our ability to detect sgRNA fragments by ~40-fold compared to lentiviral integration alone followed by traditional droplet-based scATAC-seq.

We first piloted the Spear-ATAC method with a pool of nine CRISPRi sgRNAs targeting two transcription factors (GATA1 and GATA2) and three inert sgRNA controls (Non-targeting or NT) (Supplementary Fig. 4a). We introduced this library into K562 leukemia cells engineered to express a CRISPRi dCas9-KRAB cassette to knockdown genes of interest, expanded the cells for 6 days, and then FACS-isolated sgRNA+ cells to process for Spear-ATAC. We captured 6390 nuclei in the pilot run, of which we were able to directly associate 48% of single-cell epigenetic profiles (n = 3045 nuclei) to their appropriate sgRNA target with >80% specificity (Fig. 1b and Supplementary Fig. 4b–f). Capturing the same number of cell-sgRNA assignments with existing methods would have required ~30 Perturb-ATAC runs costing $9.80/cell compared to one Spear-ATAC run costing $0.46/cell (see "Methods" and Supplementary Fig. 5). Perturb-ATAC also requires 4-hour run times on a Fluidigm C1 to process each set of 96 cells, necessitating the handling of multiple batches of frozen cells over several days. Apart from the use of standard PCR machines and bead purification steps, Spear-ATAC only requires a 7-minute run time on a 10x Controller to process up to 80,000 nuclei at once (up to 10,000 cells x 8 samples per run), greatly increasing the potential throughput of these methods (Supplementary Fig. 5).

From the 3045 nuclei assigned to sgRNAs in the pilot Spear-ATAC run, Uniform Manifold Approximation and Projection (UMAP) clearly distinguished cells harboring sgGATA1 from both sgGATA2 and sgNT cells, indicating the high specificity of sgRNA assignments (Fig. 1c and Supplementary Fig. 4c–d). GATA1 and GATA2 are both involved in hematopoietic differentiation and development; however, the erythroid transcription factor GATA1 is specifically an essential gene in K562 cells, whereas GATA2 is dispensable for growth and survival in this cell line[12]. Of note, K562 derivatives additionally have a naturally occurring side population (cluster 1 in Extended Data Fig. 4c) that has been observed and characterized in previous scATAC-seq datasets[13]. We next developed a framework for unbiased identification of changes in transcription factor (TF) motif accessibility across multiple populations of cells harboring different sgRNAs. We first computed TF motif accessibility scores (e.g., chromVAR deviation scores[14] across all single cells for a given sgRNA genotype (sgT), then subtracted the average TF motif accessibility scores of the non-targeting (sgNT) cells. We then ranked all of these sgRNA-to-TF motif accessibility difference scores (sgRNA:TF scores) to identify hits (Fig. 1d). As would be expected, knockdown of GATA1 decreased the accessibility of peaks containing the GATA motif, as well as the accessibility of peaks overlapping with known GATA1 ChIP-seq peaks (Fig. 1e). Furthermore, GATA1 knockdown resulted in a muted GATA footprint compared to K562;dCas9-KRAB cells expressing non-targeting sgRNAs (Fig. 1f). Local accessibility at the GATA1 locus also decreased following knockdown, further validating that cells assigned to sgGATA1 are down-regulating expression at this locus (Fig. 1g and Supplementary Fig. 6a).

By performing differential accessibility analysis between sgGATA1-containing cells and sgNT-expressing control cells, we observed 14,262 peaks (14.76%) increasing in accessibility and 14,026 peaks (14.52%) decreasing in accessibility (Fig. 1h). Each of the three sgRNAs targeting GATA1 resulted in nearly

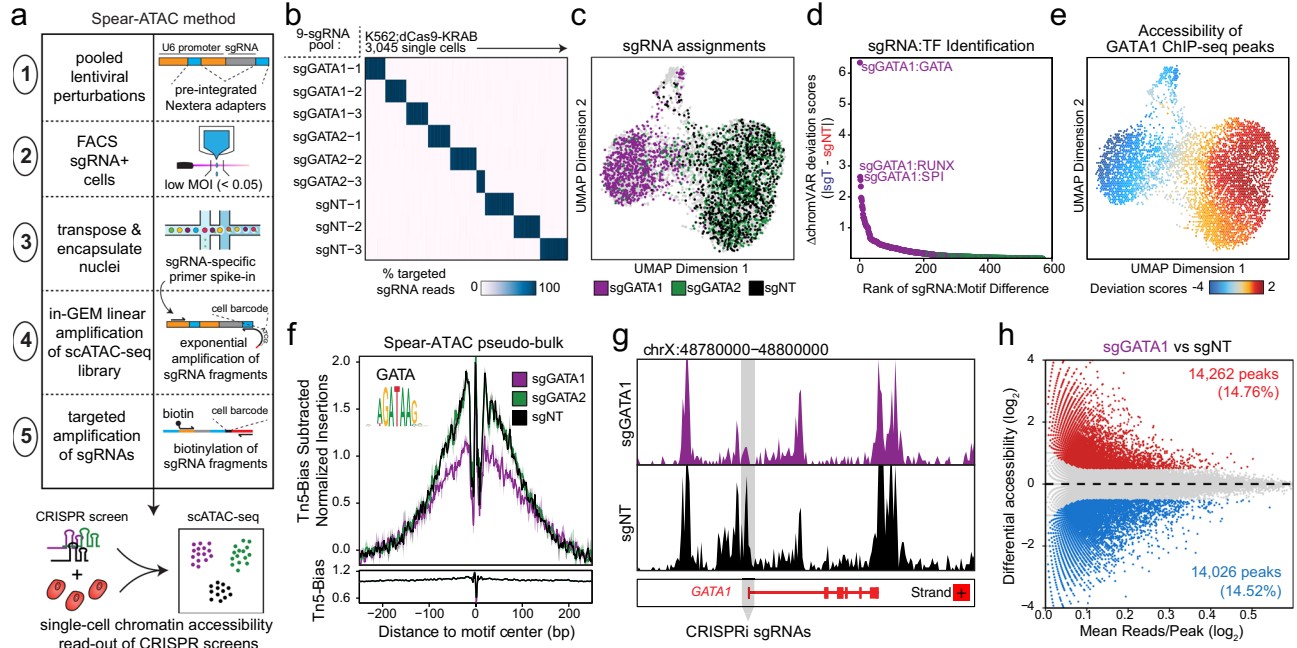

**Fig. 1 Spear-ATAC enables high-throughput CRISPR screening with a chromatin accessibility read-out. a** Schematic of the Spear-ATAC method. Modifications to traditional CRISPR screening methods and scATAC-seq approaches are outlined on the right. **b** Heatmap of the percent of sgRNA reads assigned to 3045 individual cells with corresponding chromatin accessibility information via scATAC-seq. **c** UMAP of Spear-ATAC chromatin accessibility profiles for the pilot K562 screen colored by sgRNA assignments. **d** Rank ordered plot of sgRNA:TF perturbations to identify top hits in the pilot K562 screen. **e** UMAP of Spear-ATAC chromatin accessibility profiles for the pilot K562 screen colored by chromVAR deviations for GATA1 ENCODE ChIP-seq. **f** (Top) Bias-Normalized footprint of the local accessibility for each scATAC-seq cluster for genomic regions containing GATA motifs. (Bottom) Modeled hexamer insertion bias of Tn5 around sites containing each motif. **g** Pseudo-bulk ATAC-seq track at the GATA1 locus for sgGATA1 and sgNT cells. Light grey box indicates the region targeted by sgGATA1-1, sgGATA1-2, and sgGATA1-2 CRISPRi sgRNAs. **h** Differential accessibility between sgGATA1 and sgNT cells. The x axis represents the log2 mean accessibility per peak and the y axis represents the log2 fold change in sgGATA1 cells compared to sgNT cells. Colored points are significant ($|log_2$ fold change| > 0.5, FDR < 0.05).

indistinguishable chromatin accessibility profiles (Supplementary Fig. 6b). Peaks decreasing in accessibility following GATA1 knockdown were enriched for the GATA motif and peaks increasing in accessibility following GATA1 knockdown were enriched for SPI/RUNX motifs (Supplementary Fig. 6c). Furthermore, individual cells with the lowest aggregate accessibility of genomic regions containing GATA1 motifs had the highest aggregate accessibility of genomic regions containing SPI/RUNX motifs and vice versa, further underscoring these regulatory relationships (Supplementary Fig. 6c–d). Supporting these observations, SPI (also known as PU.1) and GATA1 have been previously shown to physically interact and negatively regulate each other[15], exemplifying the type of direct phenotypes that can be assayed and validated using the Spear-ATAC method.

Beyond motif enrichment, GREAT[16] enrichment of genomic regions with decreased accessibility following GATA1 knockdown were enriched for being near erythroid-specific genes (Supplementary Fig. 6e), and genomic regions with increased accessibility following GATA1 knockdown were enriched for being near megakaryocyte-specific genes (Supplementary Fig. 6f). This result is particularly interesting given that K562 cells are often used as a model system for erythro-megakaryocytic progenitor cells. Therefore, knocking down GATA1 in K562 leukemia cells appears to prematurely push cells down a more SP1/RUNX1+, megakaryocyte lineage, despite the fact that GATA1 activity is typically required for this differentiation process[17]. Consistent with this idea, genetic disorders that impair GATA1 function often result in both the dysregulation of erythropoiesis as well as an increased incidence of transient

myeloproliferative disorder and/or acute megakaryoblastic leukemia in a subset of patients[18].

**Assessing trans-regulatory perturbations over time using Spear-ATAC.** We next took advantage of the throughput of Spear-ATAC to map the dynamic effects of knocking down transcription factors over time. Traditional proliferation based CRISPR screens evaluate the representation of sgRNAs after up to three weeks in culture; therefore, we evaluated knockdown profiles 3, 6, 9, and 21 days post-knockdown. We introduced a library of 18 sgRNAs targeting 6 transcription factors ($n = 3$ sgRNAs each) as well as 3 inert sgRNA controls into K562;dCas9-KRAB cells and performed scATAC-seq across the four time-points (Fig. 2a and Supplementary Fig. 7a-i). Similar to a proliferation-based CRISPR screen, representation of sgRNAs can be monitored over time using Spear-ATAC; for example, we observed a significant reduction in the representation of sgGATA1-containing cells at days 9 and 21 compared to days 3 and 6, whereas representation of cells with guides targeting KLF1 remained constant across days 3, 6, and 9 before decreasing at day 21 (Fig. 2b and Supplementary Fig. 7f). To identify hits from the screen—i.e., guides with significant effects on the chromatin landscape—we again used chromVAR to rank TF motif accessibility changes following sgRNA perturbations (Fig. 2c). Motif regulatory changes following GATA1 and KLF1 knockdown were the most significant across the genotypes, although the responses to sgGATA1 diminished over time corresponding to a decrease in representation of sgGATA1 cells in the population (Fig. 2c). The peaks changing in

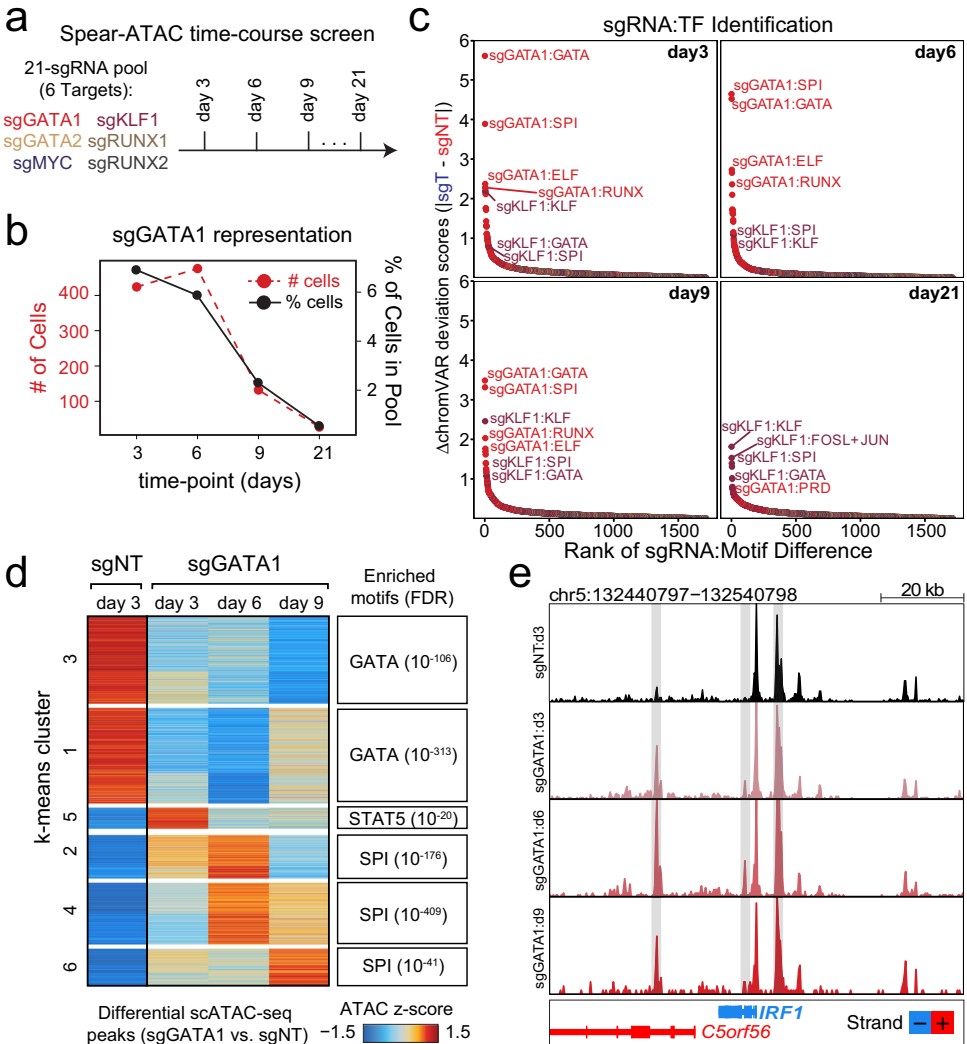

**Fig. 2 Assessing trans regulatory perturbations over time using Spear-ATAC. a** Schematic of Spear-ATAC time-course experiment with a 21-sgRNA pool analyzed at 4 time-points (3, 6, 9, and 21 days post-transduction). **b** Change in sgGATA1 representation over time, represented by the number of cells analyzed per time-point (red) and the % of cells in the total pool (black). **c** Rank ordered plot of sgRNA:TF perturbations to identify top hits in the K562 time-course screen at the indicated time points. **d** (Left) Heatmap of chromatin peak accessibility for each scATAC-seq sub-population using the top differential scATAC-seq peaks. Each row represents a z score of log$_2$ normalized accessibility within each group using scATAC-seq. Day 21 was excluded due to low representation of sgGATA1 cells at this time point. (Right) Transcription factor hypergeometric motif enrichment with FDR indicated in parentheses. **e** Pseudo-bulk ATAC-seq track at the IRF1 locus for sgGATA1 (day 3, day 6, day 9, and day 21) and sgNT cells (day3). Light grey box indicates peak regions that increased in accessibility in the sgGATA1 vs sgNT cells. Day 21 was excluded due to low representation of sgGATA1 cells at this time point.

accessibility also changed over time (Fig. 2d and Supplementary Fig. 7j); for example, peaks enriched for STAT5 motifs increased in accessibility soon after GATA1 knockdown at day 3 but returned to near baseline levels of accessibility at days 6 and 9 (Fig. 2d). STAT5 is known to be involved in the maintenance of erythroid differentiation in a GATA1-dependent process[19]; therefore, decreased accessibility of peaks containing STAT5 motifs followed by the increased accessibility of peaks containing SPI motifs at days 6 and 9 might further suggest a transition to a more megakaryocyte lineage. Local accessibility near erythroid and megakaryocytic genes also changed as a function of time following knockdown, further emphasizing the importance of timing when evaluating the effects of perturbations on chromatin accessibility (Fig. 2e and Supplementary Fig. 7i).

**High-throughput perturbation screens using Spear-ATAC.** To test the ability of Spear-ATAC to screen the chromatin

accessibility effects of transcription factors in high-throughput, we evaluated the effects of knocking down 36 transcription factors expressed in K562;dCas9-KRAB leukemia cells with 2–3 sgRNAs each, in addition to 14 control non-targeting sgRNAs and 12 sgRNAs targeting essential genes (Fig. 2a and Supplementary Fig. 8a–d). We chose a variety of transcription factors with growth effects when knocked down in K562 cells (Growth TFs) as well as ones with no proliferation phenotype following knockdown (Non-Growth TFs)[20] (Fig. 3a). Overall, we captured 32,832 nuclei representing 128 sgRNA genotypes across six Spear-ATAC samples, with on average 372 single cells being assigned to each sgRNA target with high specificity (Supplementary Fig. 8a). We next used chromVAR to rank motif accessibility changes following sgRNA perturbations and identified the top sgRNA:TF motif associations. We consistently identified the sgGATA1:GATA and sgKLF:KLF pairs as well as additional pairs such as sgNFE2:NFE2 and sgFOSL1:FOSL

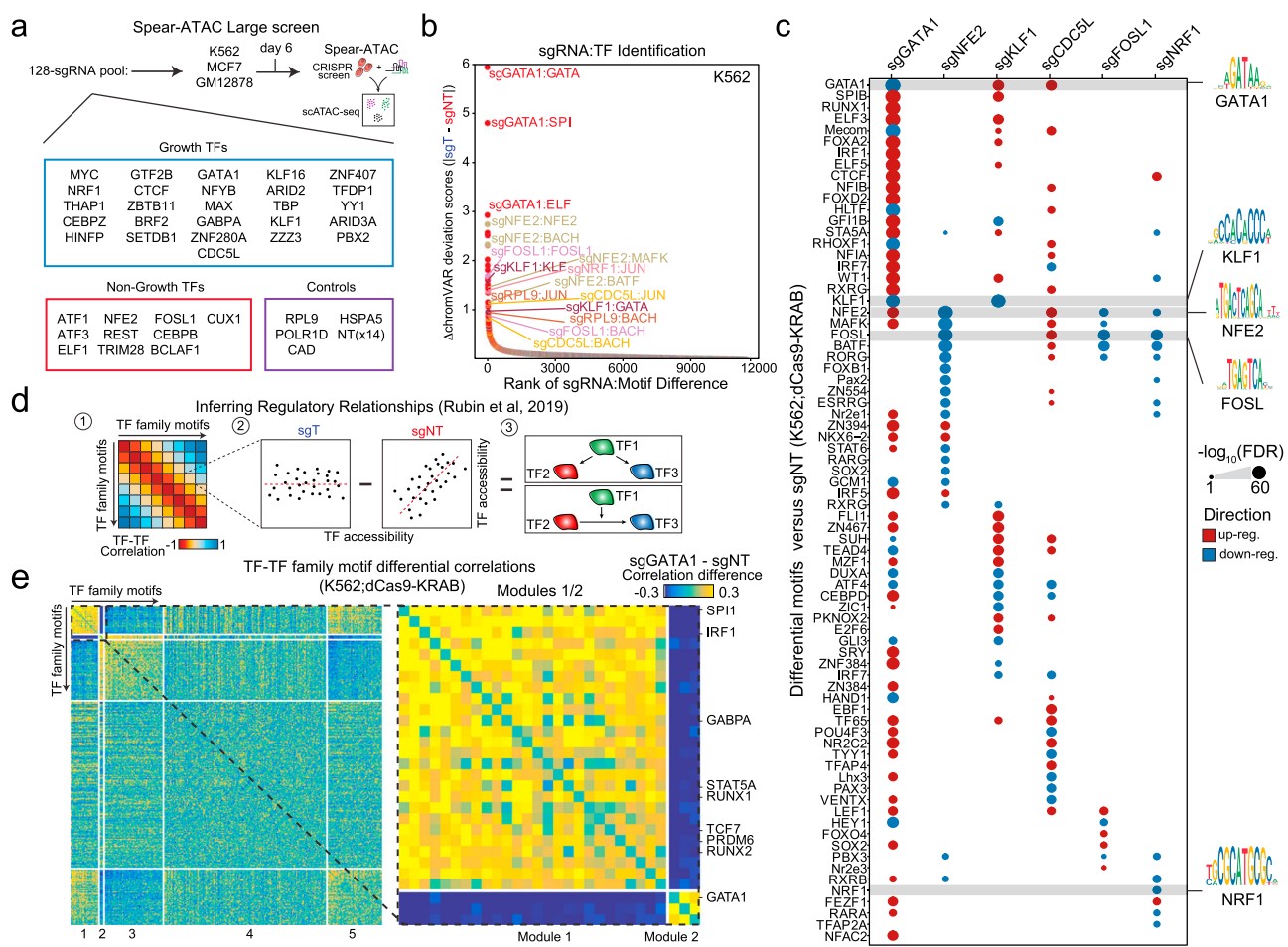

**Fig. 3 High throughput chromatin accessibility screening of CRISPR perturbations using Spear-ATAC. a** Schematic of Spear-ATAC large screens with a 132-sgRNA pool analyzed for three different cell lines (K562, MCF7, and GM12878). **b** Rank ordered plot of sgRNA:TF perturbations to identify top hits in the K562 large screen. **c** Motif enrichment in differentially accessible peaks across 6 perturbed transcription factors in the K562 large screen. Color indicates whether the motif enrichment corresponds to up-regulated (red) or down-regulated (blue) peaks. **d** Schematic of inferring regulatory relations with Spear-ATAC. Briefly, the correlation for each TF–TF motif is determined in both the targeting (sgT) and non-targeting (sgNT) cells. Next the correlation in the non-targeting cells is subtracted from targeting cells. These TF–TF motif pairs are then assessed for different regulatory relationships. **e** (Left) Heatmap of the differences between sgGATA1 targeting and non-targeting cells for all TF–TF motif accessibility correlations grouped into five different modules in the K562 large screen. (Right) Zoomed-in heatmap of modules 1 and 2 highlighting transcription factors with largely perturbed TF–TF motif accessibility relationships.

(Fig. 3b and Supplementary Fig. 9a–b). Similarly, for sgRNA genotypes that resulted in strong motif accessibility differences compared to sgNT-containing cells, the motifs identified were consistent with the targeted transcription factor, as shown for GATA1, NFE2, KLF1, FOSL1, and NRF1 (Fig. 3c).

To establish regulatory relationships between TFs, we measured the effects of TF perturbations on co-varying regulatory networks[7] (Fig. 3d). To identify these perturbed co-varying networks, we subtracted the TF–TF motif accessibility correlations within the non-targeting cells by the targeting cells. We first analyzed these relationships for the strongest target perturbation, sgGATA1. We identified five modules of TF motifs that are differentially perturbed by GATA1 knockdown (Fig. 3e). From this analysis, we again found that depletion of GATA1 led to increased coordinated activity of Module 1 consisting of crucial hematopoietic TFs such as SPI1, IRF1, RUNX, and others. To further test the specificity and performance of Spear-ATAC in additional cell lines, we performed a smaller scale screen with the same K562-optimized sgRNA targets in GM12878;dCas9-KRAB lymphoblastic cells and MCF7;dCas9-KRAB breast cancer cells. Overall, we captured an additional 12,175 cells with sgRNA

associations between the two cell lines. As expected, the sgRNA: TF motif perturbations were strongest and specific to K562 cells, highlighting cell-type specificity for TF regulation. However, shared patterns of co-varying regulatory networks were also uniquely observed in MCF7;dCas9-KRAB cells following knock-down of HINFP, CUX1, and NRF1 (Supplementary Fig. 10a–c). While HINFP, CUX1, and NRF1 have not previously been shown to directly interact with each other, HINFP and CUX1 are both involved with histone H4 gene regulation and their overlapping regulatory networks suggest a common pathway[21].

## Discussion

Spear-ATAC can be used to evaluate the effects of perturbing transcription factor expression on gene regulatory networks, increasing the throughput of previous methods by between 35- and 100-fold (depending on target nuclei capture rate) and decreasing cost by 20-fold (Supplementary Fig. 5). Of note, while this manuscript was under review, a preprint of another method to capture sgRNAs from RNA transcripts alongside scATAC-seq was made available (CRISPR-sciATAC)[22]. In contrast to both

Perturb-ATAC and Spear-ATAC, CRISPR-sciATAC relies on using 96 barcoded transposases to index ATAC and cDNA fragments from the same nucleus in plates. While the throughput of Spear-ATAC and CRISPR-sciATAC are similar, we anticipate that Spear-ATAC may be easier for laboratories to adopt due to its use of commercially available reagents.

An exciting application for Spear-ATAC will likely involve the creation of transcription factor interaction maps following multiplexed perturbations, which will enable a higher-level understanding of how proteins interact to regulate the non-coding genome. We additionally envision the application of this method following the perturbation of individual regulatory elements through high-fidelity editing methods such as prime editing[23], allowing a quantitative understanding of how disease-related mutations alter transcription factor occupancy (as inferred by ATAC-seq) and accessibility at these sites. Spear-ATAC also enables facile monitoring of pooled epigenetic perturbations across time, providing insight into the timescales involved in epigenetic reprogramming. Given the time-dependent differences we observe in our chromatin accessibility profiles following CRISPRi perturbations, we believe that this temporal dimension of monitoring is crucial for identifying the appropriate timepoint for a given study to exclude or include downstream effects. In addition, the use of inducible knockdown models with a Spear-ATAC read-out has the potential to give key insights into the mechanisms of chromatin plasticity. Finally, the Spear-ATAC workflow is not inherently limited to reading out CRISPR/Cas9 sgRNAs, but could be adapted to identify sample barcodes for higher throughput multiplexing or to read-out dynamic lineage tracing marks to understand the relationship between cells during differentiation or cancer evolution.

## Methods

**Cell lines**. Human cell lines (K562, GM12878, and MCF7) were a gift from Michael Bassik and Howard Chang's laboratories, who previously purchased them from ATCC. The dCas9-KRAB derivatives used have been validated and published previously[7,24,25]. K562;dCas9-KRAB and GM12878;dCas9-KRAB cells were cultured in RPMI media supplemented with 10% FBS, 1% penicillin-streptomycin-glutamate, and 0.1% amphotericin. MCF7;dCas9-KRAB cells were cultured in DMEM media supplemented with 10% FBS, 1% penicillin-streptomycin-glutamate, and 0.1% amphotericin. All cell lines tested negative for mycoplasma using the MycoAlert Mycoplasma Detection Kit (Lonza). Of note, while we typically think of immortalized cell lines as relatively homogeneous, K562s still exhibit and maintain natural heterogeneity, as shown by the two clusters representing the control population observed in Fig. 1c. These clusters are not unique to our dCas9-KRAB clone and we have observed that other K562-derivatives from separate sources have side populations as well. We have more extensively characterized this heterogeneity previously (Buenrostro et al. 2015). Similar side-populations have also been observed by scRNA-seq for K562s in other labs (Jost et al. 2020).

**Lentivirus production**. All lentiviruses were produced by co-transfecting lentiviral backbones with packaging vectors (delta8.2 and VSV-G) into 293 T cells using PEI (Polysciences). The virus-containing supernatant was collected at 48- and 72-h post-transfection, filtered through a 0.45 uM filter, and combined with fresh media to transduce cells. K562 and GM12878 derivatives were transduced by spinfection at 1000 g at 37 °C for 2 h. MCF7 derivatives were transduced by incubating with viral-containing supernatant for up to 2 days prior to the first fresh media change. Cell lines were incubated with 8 ug/mL polybrene (Sigma) to enhance transduction efficiency. Cells were transduced with varying amounts of virus and Spear-ATAC was only performed on cells with an MOI < 0.05 to reduce the likelihood of multiple transduction events per cell. Please see Supplementary Note 2 for a detailed protocol.

**Spear-ATAC: cloning and modifications to the sgRNA plasmid backbone**. pSP618 was derived from a modified pMJ114 backbone where the U6-sgRNA cassette was replaced with an alternate sequence from an IDT gBlock. This new U6-sgRNA cassette includes a mouse U6 promoter with a 34 bp Nextera Read2 adapter in place of the loxP site that is commonly embedded within mouse U6 promoters, followed by the original constant region (cr1) from the Perturb-seq backbones, followed by a constant region and 34 bp Nextera Read1 adapter. This plasmid will be made available on Addgene. sgRNA spacer sequences of interest were inserted into the pSP618 backbone individually using site-directed

mutagenesis (please see Supplementary Note 1 for more details). These sequences were originally picked from the Dolcetto CRISPRi genome-wide library available on Addgene and a full list is available in Supplementary Data 7. In addition, to allow for sgRNA read-out directly from sequencing the scATAC-seq library, we also cloned in unique, 10-bp sgRNA barcode sequences immediately adjacent to the Nextera Read1 adapter by site-directed mutagenesis (sequences also available in Supplementary Data 7). However, we found that targeted sgRNA amplification followed by targeted sgRNA sequencing gives the highest quality sgRNA:nuclei associations, and so we would recommend cloning in the sgRNA spacer sequences only and using the custom sequencing primer oMCB1672 (5′- GCCACTTTTTCA AGTTGATAACGGACTAGCCTTATTTTAAACTTGCTATGCTGTTTCCAGCTT AGCTCTTAAAC-3′) for Read 1 to directly sequence the sgRNA spacer sequence. sgRNA plasmids were mixed at equimolar ratios before making virus and transducing the cells of interest. Of note, for the pilot experiment, sgGATA2-3 was slightly underrepresented in the original plasmid pool for the pilot experiment, which is also reflected in the Spear-ATAC data (Fig. 1).

**Spear-ATAC: modifications to the 10x scATAC-seq protocol**. sgRNA+ nuclei were prepared for the 10x Genomics scATAC-seq protocol using version 1 of the scATAC-seq kit (10x Genomics PN-1000110)[10]. Only two modifications were necessary for Spear-ATAC. First, during GEM generation, 1.2 uL of 50 uM oSP1735 in ddH₂O (5′- gctacatttacatgataggcttgg-3′) was spiked into the Master Mix. In addition, PCR1 following GEM generation was extended from 12 cycles to 15 cycles. Please note that all primer sequences are included as part of Supplementary Data 8.

With regards to the number of nuclei to load into the 10x controller, 10x Genomics recommends that users capture between 500 and 10,000 nuclei from each sample. Loading >10,000 nuclei is not recommended by the manufacturer. Sometimes a user might be limited based on the number of nuclei available and might only choose to capture 500 nuclei total; other times the number of nuclei might not be limiting, but a user will still choose to target only 6000–7000 nuclei rather than the maximum number of 10,000 nuclei. The main downside to targeting more nuclei is that the multiplet rate (the number of gel bead emulsions that will be loaded with more than one nucleus) will increase—if a user targets 5000 nuclei, the multiplet rate is ~3.9%. If a user targets 10,000 nuclei, the multiplet rate is ~7.6% (according to the 10x Genomics website). On average in this manuscript, 6000 nuclei were targeted per sample for Spear-ATAC for K562;dCas9-KRAB samples and 4000–5000 nuclei were targeted per sample for Spear-ATAC for MCF7;dCas9-KRAB and GM12878;dCas9-KRAB samples. For the pilot screen, one sample was processed for K562;dCas9-KRAB. For the time-course screen, four samples were processed for K562;dCas9-KRAB (one for each time-point). For the large screen, six identical samples were processed in parallel for K562;dCas9-KRAB and four identical samples were processed in parallel for MCF7;dCas9-KRAB and GM12878;dCas9-KRAB samples.

**Spear-ATAC: amplification of sgRNA fragments out of the scATAC-seq libraries**. In brief, after the final scATAC-seq libraries were prepared (~150 nM final concentration in 20 uL ddH₂O), 2.5 uL of the libraries were used as input for a targeted sgRNA linear amplification PCR reaction using a 5′ biotinylated, sgRNA-specific primer (oSP2053: 5′- GTGACTGGAGTTCAGACGTGTGCTCTTCCGA TCTaagtatcccttggagaaccaccttg-3′) for 25 cycles. PCR product was pooled and purified using a Qiagen Minelute kit (Qiagen). Biotinylated sgRNA fragments were then enriched using Streptavidin MyOne C1 beads (ThermoFisher) and re-suspended in 40 uL ddH₂O, which was used as input for an exponential PCR amplification reaction for 15 cycles using primers corresponding to P5 (oSP1594: 5′- AATGATACGGCGACCACCGAGA-3′) and an indexed P7-containing primer (5′- CAAGCAGAAGACGGCATACGAGAT**NNNNNNNN**GTGACTGGAGTTC AGACGTGTG-3′, where **NNNNNNNN** is replaced with the index of choice). Please see Supplementary Note 3 for a more detailed protocol.

**Genome and transcriptome annotations**. All analyses were performed with the hg38 genome. We used the hg38 genome transcripts for gene annotations from "TxDb.Hsapiens.UCSC.hg38.knownGene".

**SpearATAC—aligning sgRNA data**. To identify the sgRNA for each single cell we first aligned each sgRNA (conventionally Read1, i.e., for 10x scATAC "R1_001. fastq.gz") to cell barcode (conventionally Index1, i.e., for 10x scATAC "R3_001. fastq.gz") combination to the sgRNA library and cell barcode library respectively. We first compiled the cell barcode library of all cell barcodes with up to 1 mismatch. We additionally created a dictionary of the sgRNA barcodes using "PDict" in R. With these 2 libraries, we read in the 2 fastq reads (Read1 and Index1) in 500,000 read chunks using the package "ShortRead" in R. We next matched the Index reads to the cell barcode library using "fmatch" in R. Then, we matched the Read1 reads to the sgRNA library using "chunkDictMatch" in R. We compiled the match results into a data frame and iterated through the full fastq reads. Finally, we created a cell by sgRNA matrix that encompassed the aligned sgRNA for each cell and identified cells that had a high-fidelity sgRNA assignment as having at least 20 sgRNA counts and a specificity of 0.8 to the top target.

**SpearATAC—aligning scATAC data**. Raw sequencing data was converted to fastq format using cellranger atac mkfastq (10x Genomics, version 1.2.0). Single-cell ATAC-seq reads were aligned to the hg38 reference genome and quantified using cellranger count (10x Genomics, version 1.2.0). The current version of Cell Ranger can be accessed here: https://support.10xgenomics.com/single-cell-atac/software/downloads/latest.

**SpearATAC—pre-processing scATAC data**. We used ArchR[26] (version 0.9.5) for all downstream scATAC-seq analysis (https://greenleaflab.github.io/ArchR_Website/). We used the fragments files for each sample with their corresponding csv file with cell information. We then created Arrow files using "createArrowFiles" with using the barcodes from the sample 10x CSV file with "getValidBarcodes". This step adds the accessible fragments a genome-wide 500-bp tile matrix and a gene-score matrix. We did not filter doublets because for these screens the cells will not form many discrete clusters and thus not many heterotypic doublets can be identified. We created an ArchRProject and then filtered cells that had a TSS enrichment below 4 and <1000 fragments. For QC plots, we used "plotGroups", "plotTSSEnrichment" and "plot-FragmentSizes". We added the sgRNA assignments for each individual sgRNA and the sgRNA targets. We reduced dimensionality with "addIterativeLSI" (default parameters), added clusters with "addClusters" (default parameters), and added a UMAP with "addUMAP" (default parameters).

To improve the fidelity of our SpearATAC sgRNA assignments, we identified the highest quality assignments for each target (similar to Replogle et al. 2020[27]). To perform this analysis, we first created an individual sgRNA by tile matrix and an sgRNA Target by tile matrix. For each target, we identified the top 5000 increasing and 5000 decreasing peaks between the target and non-targeting cells that were reproducibly regulated when comparing the individual sgRNA to the non-targeting cells. We used these 10,000 differential tile regions to perform an LSI dimensionality reduction and subsequent UMAP (n_neighbors = 40, min_dist = 0.4, metric = "cosine"). We next computed the "PurityRatio" for each sgRNA target cell based on the proportion of nearest neighbors being targeting cells ($n$ = 20). Cells that had a "PurityRatio" greater than 0.9 kept their assignment (greater than 95% of assigned cells met this criterion) for downstream analysis.

Following these assignments, we created a reproducible non-overlapping peak matrix with "addGroupCoverages" and "addReproduciblePeakSet" using the sgRNA targets as groups i.e., sgGATA1, sgGATA2, sgNT, and etc. We quantified the number of Tn5 insertions per peak per cell using "addPeakMatrix". We subsequently added motif annotations using "addMotifAnnotations" with the motifs curated and clustered from Vierstra et al. (2020)[28] (https://www.vierstra.org/resources/motif_clustering). We computed chromVAR deviations for each single cell with "addDeviationsMatrix". To identify the top sgRNA:TF perturbations, we computed the average TF motif deviations for each target and subtracted the average TF motif deviations for the non-targeting cells. By ranking the top sgRNA: TF perturbations by the absolute differences we could distinguish the top hits in each SpearATAC screen. For TF footprinting of GATA we used "plotFootprints" with normalization method "subtract" which subtracts the Tn5 bias from the ATAC footprint. When performing motif based analyses, we first ranked all motifs based on variability (relevant to the analysis) and the kept the highest motif for each motif cluster/family identified from Vierstra et al. (2020)[28] (https://www.vierstra.org/resources/motif_clustering). This filtration step removed redundant motifs, which can confound downstream analysis.

**SpearATAC—analyzing K562 pilot screen (9-sgRNA)**. Following preprocessing of the SpearATAC data, we identified differential peaks for each target vs non-target cells using "getMarkerFeatures" (testMethod = "binomial"). We identified differential peaks as those with a|log2FC| greater than 0.5 and FDR less than 0.1. Differential peaks for sgGATA1 (up-regulated and down-regulated independently) were used as input to GREAT[16] (Association = "Two nearest genes") to identify inferred regulated biological processes (i.e., GO terms). We next computed the average accessibility per peak for each individual sgRNA using "getGroupSE" (scaleTo = 10^6). To create a heatmap of differential peaks for each sgRNA of a target with sgNT (see Supplementary Fig. 6b), we subset by all differential peaks that were |log2FC| greater than 1 and then plotted a $k$ means ($k$ = 4) z score (log2-transformed) heatmap using "ArchR:::.ArchRHeatmap". To identify motifs enriched in each $k$ means cluster of peaks we used "ArchR:::.computeEnrichment" with the motifmatches and all peaks as a background set. Lastly, we computed a chromVAR deviations matrix using the ENCODE ChIP seq data set within ArchR with "addArchRAnnotations" ("EncodeTFBS") and "addDeviationsMatrix".

**SpearATAC—analyzing K562 time-course screen (21-sgRNA)**. Following preprocessing of the SpearATAC data, we identified differential peaks for each target vs non-target cells using "getMarkerFeatures" (testMethod = "binomial") for each time point (day 3, day 6, day 9, and day 21). For each time point, we identified differential peaks as those with a |log2FC| greater than 0.5 and FDR less than 0.1. We next computed the average accessibility per peak for each time point and individual sgRNA using "getGroupSE" (scaleTo = 10^6). To create a heatmap of differential peaks for each sgRNA of a target with sgNT (see Fig. 2d and Supplementary Fig. 7j), we first subset by the union of all differential peaks that were |log2FC| greater than 1 for each time point. Next, we computed the average log2 fold changes for sgRNA target vs the sgNT at that time point (using the pseudobulk

matrix above). We further filtered the differential peaks by those peaks that have a | log2FC| greater than 0.25 in at least 1 time point. We plotted a $k$ means ($k$ = 6) z-score (log2-transformed) heatmap using "ArchR:::.ArchRHeatmap". To identify motifs enriched in each $k$ means cluster of peaks we used "ArchR:::.computeEnrichment" with the motifmatches and all peaks as a background set.

**SpearATAC—analyzing large screens for K562, GM12878, and MCF7 (128-sgRNA)**. Following preprocessing of the SpearATAC data, we identified differential motifs for each target vs non-target cells using "getMarkerFeatures" (testMethod = "wilcoxon", bufferRatio = 0.95, maxCells = 250, useSeqnames = "z"). We filtered sgNT-5,6,8,11,12 cells prior to this differential comparison after identifying these sgRNA as outliers while performing pseudobulk PCA analysis. To identify perturbed co-varying regulatory networks (see Fig. 3e and Supplementary Fig. 10a–c), we first got the motif deviations matrix for each screen (K562, GM12878, and MCF7) and filtered cells corresponding to sgNT-5,6,8,11,12. Next, we computed the average motif deviation scores for each sgRNA target. We subtracted the average motif deviation scores from the non-targeting cells. We rank ordered the motifs by the maximum observed average motif deviation score difference (absolute difference) across all targets in each screen. Finally, we rank ordered the motifs by the average of these maxima across all three screens (K562, GM12878, and MCF7). We de-duplicated the motifs in each cluster (see Vierstra et al. 2020[28]; https://www.vierstra.org/resources/motif_clustering) to remove redundant motifs. For each sgRNA we computed all TF–TF deviation score correlations for the non-redundant motifs. Each targeting sgRNA was then subtracted by the non-targeting sgRNA TF–TF correlations. This differential correlation matrix was subsequently hierarchal clustered with "hclust" and split into five modules with "cutree". A heatmap of the differential correlations for the sgRNA targeting cells was then constructed across all modules.

**Spear-ATAC—cost analysis vs C1 fluidigm perturb-ATAC (Rubin et al., 2019)**. To compare the cost of capturing both scATAC and sgRNA in the same cell for Spear-ATAC vs Perturb-ATAC we first calculated the cost for capturing scATAC for each method respectively (~$750 for C1 Fluidigm scATAC for ~96 cells and ~ $1400 for 10x scATAC for 500–10,000 cells). We ignored the cost of sequencing in this analysis because previously published results differ in the total reads sequenced per cell (C1 Fluidigm experiments have traditionally been over sequenced because of lower complexity scATAC libraries vs 10x scATAC libraries)[10,29]. We next determined the sgRNA capture for both methods (85% for Perturb-ATAC and 48% in Spear-ATAC for the K562 Pilot experiment). We additionally simulated a range of capture rates for Spear-ATAC to account for capture rate variability. We used these values to then determine the cost per cell with both scATAC and sgRNA (Cost Per Sample / Number of cells with both scATAC and sgRNA). With this estimation we determined the total cost given the number of cells desired and the number of kits needed to generate this cell count. We note that each 10x kit has eight samples that can be performed in parallel for massive throughput. These numbers should serve as a reference for future adoption of the Spear-ATAC method.

**Reporting summary**. Further information on research design is available in the Nature Research Reporting Summary linked to this article.

## Data availability

All matrices (peak matrix and chromVAR) are available through AWS (see Supplementary Data 6). We also made the 10x cell ranger atac output files and all scATAC-seq matrices used in this study available through AWS (see Supplementary Data 6). All sequencing data have been deposited in the Gene Expression Omnibus (GEO) at [GSE168851]. Plasmids generated in this study are available from the Lead Contact without restriction. The Spear-ATAC lentiviral backbone (pSP618) is available on Addgene and the sequence is on GenBank with accession MW852482. Any other relevant data are available from the authors upon reasonable request. Source data are provided with this paper.

## Code availability

All custom code used in this work is available upon request. We additionally are hosting a Github website that includes the main analysis code used in this study as well as a tutorial with sample data (https://github.com/GreenleafLab/SpearATAC_MS_2021)[30].

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

## Acknowledgements

We thank M. Bassik, Josh Tycko, and members of the Greenleaf, Winslow, and Chang laboratories for helpful comments. We thank the Stanford Shared FACS facility for technical support and A. Orantes for administrative support. Funding: S.E.P was supported by the NSF Graduate Research Fellowship Award and the Tobacco-Related Diseases Research Program Predoctoral Fellowship Award. This work was supported by NIH RM1-HG007735, UM1-HG009442, UM1-HG009436, R01-HG00990901, and U19-AI057266 (to W.J.G.). W.J.G. acknowledges funding from Emmerson Collective. W.J.G. is a Chan-Zuckerberg investigator.

## Author contributions

S.E.P., J.M.G., and W.J.G conceived the project and designed the experiments. S.E.P. led the method development and experimental data production. J.M.G led the data analysis. J.M.G. and S.E.P. performed the ATAC-seq and scATAC-seq analysis. S.E.P. and J.M.G were supervised by W.J.G. and S.E.P., J.M.G., and W.J.G wrote the paper.

## Competing interests

The authors declare the following competing interests: W.J.G. is a consultant for 10x Genomics who has licensed IP associated with ATAC-seq. W.J.G. has additional affiliations with Guardant Health (consultant) and Protillion Biosciences (co-founder and consultant). The remaining authors declare no competing interests.
