## [Peer Review File · Nature Communications]

Reviewers' Comments:

Reviewer #1:

Remarks to the Author:

Summary: Pierce et al. describe a droplet based CRISPR pooled screening system paired with ATAC-seq. The design is clever in that gDNA (instead of mRNA) is amplified along with the ATAC-seq method and further biotin captured. We find this article to be a modest improvement over existing methodology (Rubin et al. 2019). The article's major significance is that the technique is more cost effective (the authors claim close to 20x cheaper). The authors claim that the method is high throughput and that the technique allows for temporal monitoring of chromatin accessibility changes. Given clarification of the text and addressing the below concerns, this study would be appropriate for publication in Nature Communications.

Claims: Do the authors provide appropriate evidence to support their claims?

Claim 1: Spear-ATAC is more cost effective than existing technology.

The authors demonstrate a compelling cost benefit.

Claim 2: Spear-ATAC enables high throughput CRISPR screening.

The case can be made that the scale of the experiments carried out here do not reach a "high throughput" threshold. The authors would be best served by using "medium throughput" terminology or scale experiments up to >1,000 sgRNAs.

Claim 3: Temporal monitoring of chromatin accessibility changes with TF repression.

To demonstrate the applicability of this method, the authors should scale up the number of target genes with temporal monitoring (only 6 genes assayed in Figure 2). In Figure 3 the authors screen 38 genes, yet do not produce temporal data here. Also, the authors might consider employing an inducible dCas9-KRAB to rescue accessibility over the time course to demonstrate temporal changes.

Specific Points

Page 4, Line 63-65: "We captured 6,390 nuclei in the pilot run, of which we were able to directly associate 48% of single-cell epigenetic profiles (n=3,045 nuclei) to their appropriate sgRNA target with >80% specificity (Fig 1b and Extended Data Fig. 4b-e)."

- How efficiently does this technology detect sgRNA in the infected single cells compared to the previous technology (Rubin, A. J. et al., Cell 2019)?
- Why is the sgGATA2-3 population much smaller than others (Fig 1b)?

Page 5, Line 74-77: "From the 3,045 nuclei assigned to sgRNAs in the pilot Spear-ATAC run, Uniform Manifold Approximation and Projection (UMAP) clearly distinguished cells harboring sgGATA1 from both sgGATA2 and sgNT cells, indicating the high specificity of sgRNA assignments (Fig. 1c and Extended Data Fig. 4c). GATA1 and GATA2 are both involved in hematopoietic differentiation and development; however, the erythroid transcription factor GATA1 is specifically an essential gene in K562 cells, whereas GATA2 is dispensable for growth and survival in this cell line12."

- UMAP appears to consist of 2 major clusters and 1 minor cluster on Fig. 1c. On the extended data Fig. 4c, Seurat graph clustering in ArchR shows a single cluster for sgGATA1 and multiple clusters for sgGATA2 and sgNT cells. Why are there multiple clusters found in the control population? Authors should address in text.
- GATA2 repression does not decrease GATA accessibility (Fig. 1c, 1f and 2c). How is this explained? Have the authors confirmed repression of GATA2 transcripts?

Page 6, Line 95-97: "Interestingly, knocking down GATA1 led to a modest increase in accessibility of GATA3 ChIP-seq peaks as well as an increase in local accessibility near the GATA3 locus (Extended Data Fig. 5b-c). GATA3 is typically active in the lymphoid lineage 14 and is not expressed in K562 cells at baseline, suggesting that GATA3 is specifically activated in response to

GATA1 knock down.”

- This is speculative. Authors demonstrated that the chromatin accessibility near GATA3 locus was changed by GATA1 knock down (Extended Data Fig. 5b-c). However, there is no direct evidence for the specific activation of GATA3 gene in response to GATA1 knock down. Have the authors confirmed the change of GATA gene expression?

Page 7, Line 125-132: “We next took advantage of the throughput of Spear-ATAC to map the dynamic effects of knocking down transcription factors over time. Traditional proliferation based CRISPR screens evaluate the representation of sgRNAs after up to three weeks in culture; therefore, we evaluated knockdown profiles 3, 6, 9, and 21 days postknockdown. We introduced a library of 18 sgRNAs targeting 6 transcription factors (n=3 sgRNAs each) as well as 3 inert sgRNA controls into K562;dCas9-KRAB cells and performed scATAC-seq across the four time-points (Fig. 2a and Extended Data Fig. 6a-h).”

- When authors demonstrated the ability of Spear-ATAC to reveal the dynamic chromatin accessibility changes of TF knock down over time, they used a library of only 18 sgRNAs targeting 6 transcription factors (Fig. 2). Can this technology be scaled up for high throughput CRISPR screening (>1000 genes)?

Page 8, Line 142-143: “The peaks changing in accessibility also changed over time (Fig. 2d and Extended Data Fig. 6j)”

Page 27, Line 587-589: “e. Pseudo-bulk ATAC-seq track at the IRF1 locus for sgGATA1 (day3, day6, day9, and day21) and sgNT cells (day3). Light grey box indicates peak regions that increased in accessibility in the sgGATA1 vs sgNT cells.”

Page 33, Line 724-727: “i. Pseudo-bulk ATAC-seq track at the (Top Left) RUNX1, (Top Right) PRKAR2B, (Bottom Left) PPBP and (Bottom Right) MPL locus for sgGATA1 (day3, day6, day9, and day21) and sgNT cells (day3). Light grey box indicates peak regions that changed in accessibility in the sgGATA1 vs sgNT cells.”

- Where is Day21 data (Fig. 2d-e, Extended Data Fig. 6j-i)? Authors need to display the Day21 data or explain why the data was excluded.

Page 8, Line 150-153: “Local accessibility near erythroid and megakaryocytic genes also changed as a function of time following knockdown, further emphasizing the importance of timing when evaluating the effects of perturbations on chromatin accessibility (Fig. 2e and Extended Data Fig. 6i).”

- Authors used pseudo-bulk ATAC-seq with single-cell epigenetic profiles on Fig. 2e and Extended Data Fig. 6i. Is it unlikely that such temporal changes are due to the average measurement of chromatin accessibility across a population of single cells?

Page 8, Line 154-158: “Finally, to test the ability of Spear-ATAC to screen the chromatin accessibility effect of transcription factors in high-throughput, we evaluated the effects of knocking down 38 transcription factors expressed in K562;dCas9-KRAB leukemia cells with 2-3 sgRNAs each, in addition to 15 control non-targeting sgRNAs and 16 sgRNAs targeting essential genes (Fig. 3a and Extended Data Fig. 7a-d).”

- Authors addressed 38 TFs genes in text. But 35 TFs (Growth + Non-growth) are listed on Fig. 3a. and 36 TFs in supplementary table 7. In addition, they addressed that 15 control non-targeting sgRNAs and 16 sgRNAs targeting essential genes were used for Spear-ATAC on Fig. 3a. But, in supplementary table 7, the numbers are 14 for non-targeting sgRNAs and 12 for sgRNAs targeting essential genes. Authors should check and correct the number of targeting genes.

- CDC5L is not shown on Fig. 3a but in Fig. 3b-c. Authors need to add CDC5L.

Page 8, Line 161-163: “Overall, we captured 32,832 nuclei representing 128 sgRNA genotypes across six Spear-ATAC samples, with on average 372 single cells being assigned to each sgRNA target with high specificity (Extended Data Fig. 7a).”

- Why GM12878 and MCF7 showed relatively poor numbers of nuclei with high sgRNA assignment specificity, comparing with K562?

Page 9, Line 167-170: "Similarly, for sgRNA genotypes that resulted in strong motif accessibility differences compared to sgNT-containing cells, the motifs identified were often consistent with the targeted transcription factor, as shown for GATA1, NFE2, KLF1, FOSL1, and NRF1 (Fig. 3c)."

- What kinds of the motifs identified were consistent or inconsistent with the targeted TFs? Also, authors should explain why such a difference occurred.

Reviewer #2:

Remarks to the Author:

The authors describe a method to quantify and map the effects of perturbing transcription factor levels on chromatin accessibility in high throughput. Spear-ATAC (Single-cell perturbations with an accessibility read-out using scATAC-seq) is a droplet-based single-cell ATAC-seq protocol that enables simultaneous read-out of chromatin accessibility profiles and integrated sgRNA spacer sequences from thousands of individual cells at a time. It relies on reading out sgRNA spacer sequences directly from gDNA rather than from RNA transcripts.

As the authors discuss, this is not the first time that a method combining CRISPR screening with chromatin accessibility profiling of single cells is described. The previously published Pertub-ATAC (Rubin et al., 2019, Cell) is limited to analyzing 96 cells per run on a microfluid device. Spear-ATAC promises to increase the screening throughput by at least 35-fold and to decrease the experimental cost by 20-fold. Also the fact that it is based on 10x genomics technology for capturing single nuclei, makes this protocol attractive for many labs that are currently using 10x genomics for scATAC-seq analysis.

Given that Spear-ATAC protocol may be used by labs with no strong background in developing such protocols (but rather rely on manuscripts like the one under review) it is important that the authors put some more effort in describing the protocol in more details. Fig 1a and extended Data Fig. 1a is a good start, but not very detailed. Especially more details should be given on the sgRNA plasmid. The authors provide a detailed protocol only for the targeted amplification of sgRNAs (Supplementary Note 1). It would be important that they do so for all steps of Spear-ATAC (from cloning and virus production to data analysis). The same applies to code availability. The authors promise to provide custom code used upon request, but they may want to make it available via Github etc.

There is some misleading regarding the number of cells profiled by Spear-ATAC. In the abstract it is mentioned that 104,592 cells (representing 414 sgRNAs) were profiled, giving the impression that this cell number comes from a single Spear-ATAC run. In a similar manner the authors comment in the manuscript that Spear-ATAC allows to profile up to 80,000 nuclei at once (line 72). However, the highest number of cells processed at once in this manuscript is 32,832. Is this correct? The authors may want to clarify this issue. What is the number of fragments sequenced per cells assigned to each sgRNA?

Line 97: The authors suggest that GATA3 is activated upon GATA1 kd but this is not shown at the expression level. An ATAC-seq peak in the gene body of GATA3 after GATA1 kd does not necessarily mean increase in the expression levels of the corresponding gene.

While this manuscript was under review, a manuscript with a similar focus was posted on bioRxiv (Liscovitch-Brauer et al) <https://www.biorxiv.org/content/10.1101/2020.11.20.390971v1> . The authors may comment on this method and how it compares to Spear-ATAC in the discussion.

Point-by-point response
NCOMMS-20-43842-T

We thank the Reviewers for their thoughtful assessment and critique of the manuscript, which have helped us significantly improve this work. Overall, the manuscript has now been revised to include several new panels and textual changes, as suggested by the reviewers. Please find our detailed responses to the suggestions in blue below and **highlighted within the main text file**. Page numbers of added text and Figure call-outs are underlined in blue in the response.

A summary of the major additions to the manuscript is provided below:

1. Detailed protocols for every step of Spear-ATAC, from cloning to virus prep to computational analysis.

a. Additional Supplementary Notes with detailed protocol information.

Reviewer 2 asked for additional detail outlining how to perform this method, particularly for laboratories that might not be accustomed to making CRISPR libraries or doing this type of work. We have now included several Supplementary Notes outlining these steps.

b. Code deposited on Github with a new tutorial available.

Reviewer 2 also asked for additional information on how to analyze this type of data. We now provide code deposited on Github (https://github.com/GreenleafLab/SpearATAC_MS_2021) as well as a tutorial with sample data included to learn how to analyze Spear-ATAC results.

2. Revised Figures and text for clarity.

a. Clarifications of the number of cells that can be processed with Spear-ATAC.

Reviewers 1 and 2 had clarifying questions about the number of cells that can be processed with Spear-ATAC and the number of cells that can be assigned to sgRNAs compared to previous methods. We have now included a supplemental figure showing the cost of Spear-ATAC compared to previous methods depending on the cell capture rate and number of cells appropriately assigned to sgRNAs. We have also clarified the numbers throughout the text to more clearly explain how they are being derived.

b. Responses to additional clarifying questions brought up in Review.

Reviewer 1 had many detailed questions related to clarifying information in specific panels as well as further details of our method. We have now added the appropriate information within the text, figure legends, and/or Methods section as appropriate.

3. Title change.

We changed the title to “Spear-ATAC: Pooled, droplet-based, single-cell chromatin accessibility CRISPR screens” to emphasize the method.

REVIEWER COMMENTS

Reviewer #1 (Remarks to the Author):

Summary: Pierce et al. describe a droplet based CRISPR pooled screening system paired with ATAC-seq. The design is clever in that gDNA (instead of mRNA) is amplified along with the ATAC-seq method and further biotin captured. We find this article to be a modest improvement over existing methodology (Rubin et al. 2019). The article's major significance is that the technique is more cost effective (the authors claim close to 20x cheaper). The authors claim that the method is high throughput and that the technique allows for temporal monitoring of chromatin accessibility changes. Given clarification of the text and addressing the below concerns, this study would be appropriate for publication in Nature Communications.

We thank the Reviewer for their high level of detail when reviewing our manuscript and we appreciate how much it has improved the clarity of the text and the readability for future users of Spear-ATAC.

Claims: Do the authors provide appropriate evidence to support their claims?

Claim 1: Spear-ATAC is more cost effective than existing technology.

The authors demonstrate a compelling cost benefit.

We thank the Reviewer for their positive assessment of the cost benefit of Spear-ATAC.

Claim 2: Spear-ATAC enables high throughput CRISPR screening.

The case can be made that the scale of the experiments carried out here do not reach a "high throughput" threshold. The authors would be best served by using "medium throughput" terminology or scale experiments up to >1,000 sgRNAs.

We thank the Reviewer for trying to establish an appropriate reference for what qualifies as "high-throughput". This point is especially important given that the throughput of single-cell technologies is increasing rapidly. However, the single-cell field has already labeled equivalent single-cell RNA-based methods as high-throughput after targeting even fewer genes; for example, the largest CROP-seq screen in the original paper involved targeting 29 genes (Datlinger et al. 2017) and the largest Perturb-seq screen in the original paper involved targeting 24 genes (Dixit et al. 2016). We target 36 transcription factors and 4 essential genes (40 genes total) in our largest Spear-ATAC screen. In addition, the original Fluidigm-based Perturb-ATAC method was labeled as high-throughput after assaying only 4,300 cells (Rubin et al. 2019). We assay >100,000 cells with Spear-ATAC. Therefore, while we understand the intention of the Reviewer in making this comment, we do not think saying that Spear-ATAC is "medium throughput" would be consistent with what has already been established in this field.

Claim 3: Temporal monitoring of chromatin accessibility changes with TF repression.

To demonstrate the applicability of this method, the authors should scale up the number of target genes with temporal monitoring (only 6 genes assayed in Figure 2). In Figure 3 the authors screen 38 genes, yet do not produce temporal data here. Also, the authors might consider employing an inducible dCas9-KRAB to rescue accessibility over the time course to demonstrate temporal changes.

We thank the Reviewer for this suggestion to increase the number of sgRNAs assayed by temporal monitoring. We are looking forward to additional users of our method assaying numerous knockdown phenotypes over time using Spear-ATAC. However, at this time we do

Point-by-point response NCOMMS-20-43842-T

not think that increasing the number of sgRNAs would further demonstrate the proof-of-principle applicability of our method. In particular, as the Reviewer points out, we follow up the time-course experiment with a screen of 40 genes (128 sgRNAs total). Therefore, we have shown that it is possible to both use Spear-ATAC to monitor changes in chromatin state over time and also to scale up Spear-ATAC to monitor >100 sgRNA genotypes. Even more importantly, while Spear-ATAC greatly reduces the cost of this type of method compared to Perturb-ATAC, repeating the 128-sgRNA screen over four time points would still cost (at minimum) >\$8,000 per cell line (\$2,100 per 10x sample x at least one sample per time-point). Therefore, we think that specific biological questions would be necessary to justify the cost of repeating this experiment.

We thank the Reviewer for the suggestion to employ an inducible dCas9-KRAB to study the temporal nature of accessibility changes. We have now added a new sentence to our Discussion section proposing this type of inducible model for future Spear-ATAC experiments (page 11, lines 214-216):

“In addition, the use of inducible knockdown models with a Spear-ATAC read-out has the potential to give key insights into the mechanisms of chromatin plasticity.”

Specific Points

Page 4, Line 63-65: “We captured 6,390 nuclei in the pilot run, of which we were able to directly associate 48% of single-cell epigenetic profiles (n=3,045 nuclei) to their appropriate sgRNA target with >80% specificity (Fig 1b and Extended Data Fig. 4b-e).”

- How efficiently does this technology detect sgRNA in the infected single cells compared to the previous technology (Rubin, A. J. et al., Cell 2019)?

Spear-ATAC can detect sgRNAs in up to 48% of cells whereas Perturb-ATAC can detect sgRNAs in up to ~85% of cells (Rubin et al. 2019). This difference is expected given that Spear-ATAC is detecting sgRNAs off of a single copy of DNA whereas Perturb-ATAC is detecting sgRNAs off of many copies of RNA. To further clarify and emphasize the differences between these methods, we have now added a new supplementary figure to show the throughput and comparisons of Spear-ATAC and Perturb-ATAC given various cell and sgRNA capture rates (Supplementary Fig. 5).

- Why is the sgGATA2-3 population much smaller than others (Fig 1b)?

On average each sgRNA is approximately equally represented in the pool. While sgGATA2-3 appears slightly underrepresented compared to the other sgRNAs, this underrepresentation is reflected in the plasmid pool and therefore is not due to off-target effects of the sgRNA that altered the proliferation of K562;dCas9-KRAB cells. Instead, this likely reflects a small error in the manual pooling of the original plasmids. We have now indicated this point in the Methods to avoid any future confusion (page 15, lines 310-312):

“Of note, for the pilot experiment, sgGATA2-3 was slightly underrepresented in the original plasmid pool for the pilot experiment, which is also reflected in the Spear-ATAC data (Figure 1).”

Page 5, Line 74-77: “From the 3,045 nuclei assigned to sgRNAs in the pilot Spear-ATAC run, Uniform Manifold Approximation and Projection (UMAP) clearly distinguished cells harboring sgGATA1 from both sgGATA2 and sgNT cells, indicating the high specificity of sgRNA assignments (Fig. 1c and Extended Data Fig. 4c). GATA1 and GATA2 are both involved in hematopoietic differentiation and development; however, the erythroid transcription factor GATA1 is specifically an essential gene in K562 cells, whereas GATA2 is dispensable for

Point-by-point response NCOMMS-20-43842-T

growth and survival in this cell line12.”

- UMAP appears to consist of 2 major clusters and 1 minor cluster on Fig. 1c. On the extended data Fig. 4c, Seurat graph clustering in ArchR shows a single cluster for sgGATA1 and multiple clusters for sgGATA2 and sgNT cells. Why are there multiple clusters found in the control population? Authors should address in text.

While we typically think of immortalized cell lines as relatively homogeneous, K562s still exhibit and maintain natural heterogeneity, as shown by the two clusters representing the control population observed in Figure 1c. These clusters are not unique to our dCas9-KRAB clone and we have observed that other K562-derivatives from separate sources have distinct sub-populations as well. We have more extensively characterized this heterogeneity previously (Buenrostro et al. 2015). Side-populations have also been observed by scRNA-seq for K562s in other labs (Jost et al. 2020) and we also note that MCF7s and GM12878 cells similarly have naturally occurring side-populations (see Supplementary Figure 8d). We have now noted this information in the main text (page 5, lines 82-85):

“Of note, K562 derivatives additionally have a naturally occurring side population (cluster 1 in Extended Data Fig. 4c) that has been observed and characterized in previous scATAC-seq datasets (Buenrostro et al. 2015).”

We have also noted this information in our Methods section under *Cell lines* (page 13, lines 267-274):

“Of note, while we typically think of immortalized cell lines as relatively homogeneous, K562s still exhibit and maintain natural heterogeneity, as shown by the two clusters representing the control population observed in Figure 1c. These clusters are not unique to our dCas9-KRAB clone and we have observed that other K562-derivatives from separate sources have side populations as well. We have more extensively characterized this heterogeneity previously (Buenrostro et al. 2015). Similar side-populations have also been observed by scRNA-seq for K562s in other labs (Jost et al. 2020).”

- GATA2 repression does not decrease GATA accessibility (Fig. 1c, 1f and 2c). How is this explained? Have the authors confirmed repression of GATA2 transcripts?

We thank the Reviewer for this question, and we have now included qPCR for GATA2 quantifying the appropriate on-target repression of GATA2 transcripts (Supplementary Fig. 4f), leading to a >50% knockdown of GATA2 mRNA transcripts. While GATA2 has a similar binding motif as GATA1, that does not inherently mean that GATA2 maintains accessibility at these sites and we do not observe any significant differences in GATA motif accessibility following GATA2 knockdown. In fact, including GATA2 in the pilot experiment was originally intended as a control for GATA1 knockdown; in contrast to GATA1, GATA2 is not an essential gene in K562 cells and has no proliferation effect when knocked down in this setting. Therefore, having no significant effect on chromatin accessibility was not surprising. This information is included in the manuscript text as well (page 5, lines 80-83):

Point-by-point response NCOMMS-20-43842-T

“GATA1 and GATA2 are both involved in hematopoietic differentiation and development; however, the erythroid transcription factor GATA1 is specifically an essential gene in K562 cells, whereas GATA2 is dispensable for growth and survival in this cell line.”

Page 6, Line 95-97: “Interestingly, knocking down GATA1 led to a modest increase in accessibility of GATA3 ChIP-seq peaks as well as an increase in local accessibility near the GATA3 locus (Extended Data Fig. 5b-c). GATA3 is typically active in the lymphoid lineage 14 and is not expressed in K562 cells at baseline, suggesting that GATA3 is specifically activated in response to GATA1 knock down.”

- This is speculative. Authors demonstrated that the chromatin accessibility near GATA3 locus was changed by GATA1 knock down (Extended Data Fig. 5b-c). However, there is no direct evidence for the specific activation of GATA3 gene in response to GATA1 knock down. Have the authors confirmed the change of GATA gene expression?

Upon reflection, we agree with the Reviewer that our initial wording was too strong given the data we had collected at the time. We attempted to carry out qPCR for GATA3 following GATA1 knockdown, but it appears that GATA3 is either not expressed or expressed at very low levels in GATA1-knockdown cells. We have now taken out the two panels relating to this observation to avoid future confusion.

Page 7, Line 125-132: “We next took advantage of the throughput of Spear-ATAC to map the dynamic effects of knocking down transcription factors over time. Traditional proliferation based CRISPR screens evaluate the representation of sgRNAs after up to three weeks in culture; therefore, we evaluated knockdown profiles 3, 6, 9, and 21 days postknockdown. We introduced a library of 18 sgRNAs targeting 6 transcription factors (n=3 sgRNAs each) as well as 3 inert sgRNA controls into K562;dCas9-KRAB cells and performed scATAC-seq across the four time-points (Fig. 2a and Extended Data Fig. 6a-h).”

- When authors demonstrated the ability of Spear-ATAC to reveal the dynamic chromatin accessibility changes of TF knock down over time, they used a library of only 18 sgRNAs targeting 6 transcription factors (Fig. 2). Can this technology be scaled up for high throughput CRISPR screening (>1000 genes)?

The Spear-ATAC protocol does not have any dependency on the number of genes targeted – the protocol for capturing sgRNAs is identical whether a user targets 10 genes or 1000 genes. In contrast to Perturb-ATAC, throughput is substantially less of a limiting factor for Spear-ATAC, as each 10x Controller run can technically process up to 80,000 nuclei at once, while the Fluidigm chip used in Perturb-ATAC can assay 96 cells maximally at once. In some sense, therefore, the major limitation to this method is the amount that a user is willing to spend on reagents, which as we have discussed, is approximately 20-fold less for Spear-ATAC than Perturb-ATAC. We have now included a supplementary figure to show how the cost and throughput changes depending on the number of cells captured per sample and the percent of cells associated to a sgRNA (Supplementary Figure 5).

Page 8, Line 142-143: “The peaks changing in accessibility also changed over time (Fig. 2d and Extended Data Fig. 6j)”

Page 27, Line 587-589: “e. Pseudo-bulk ATAC-seq track at the IRF1 locus for sgGATA1 (day3, day6, day9, and day21) and sgNT cells (day3). Light grey box indicates peak regions that increased in accessibility in the sgGATA1 vs sgNT cells.”

Page 33, Line 724-727: “i. Pseudo-bulk ATAC-seq track at the (Top Left) RUNX1, (Top Right) PRKAR2B, (Bottom Left) PPBP and (Bottom Right) MPL locus for sgGATA1 (day3, day6, day9,

Point-by-point response NCOMMS-20-43842-T

and day21) and sgNT cells (day3). Light grey box indicates peak regions that changed in accessibility in the sgGATA1 vs sgNT cells.”

- Where is Day21 data (Fig. 2d-e, Extended Data Fig. 6j-i)? Authors need to display the Day21 data or explain why the data was excluded.

Day 21 data was not included because GATA1 is an essential gene in K562s (Fulco et al. 2016) and the representation of sgGATA1-containing cells was extremely low at day 21 in the pool (see Figure 2b). We thank the Reviewer for bringing this to our attention and have clarified in the figure legends why this data was not shown to avoid future confusion.

Page 8, Line 150-153: “Local accessibility near erythroid and megakaryocytic genes also changed as a function of time following knockdown, further emphasizing the importance of timing when evaluating the effects of perturbations on chromatin accessibility (Fig. 2e and Extended Data Fig. 6i).”

- Authors used pseudo-bulk ATAC-seq with single-cell epigenetic profiles on Fig. 2e and Extended Data Fig. 6i. Is it unlikely that such temporal changes are due to the average measurement of chromatin accessibility across a population of single cells?

We thank the Reviewer for this comment. All differentially accessible regions were first identified using clusters of single cells in ArchR, which takes into account single-cell variability by treating each single-cell as its own replicate in each cluster and therefore is not the same as a fully bulk measurement. In addition, ArchR accounts for experimental bias by matching cells with a similar TSS/number of fragments per cell in each cluster before identifying differential peaks. We then compile those differentially accessible regions and show these regions as pseudobulks in a heatmap (Figure 2d) to more easily visualize the variation of hundreds to thousands of single cells across thousands of regions. However, since these regions were identified by performing differential analysis at the single-cell level, we do not have a reason to believe that these differences are due to the averaging of measurements.

Page 8, Line 154-158: “Finally, to test the ability of Spear-ATAC to screen the chromatin accessibility effect of transcription factors in high-throughput, we evaluated the effects of knocking down 38 transcription factors expressed in K562;dCas9-KRAB leukemia cells with 2-3 sgRNAs each, in addition to 15 control non-targeting sgRNAs and 16 sgRNAs targeting essential genes (Fig. 3a and Extended Data Fig. 7a-d).”

- Authors addressed 38 TFs genes in text. But 35 TFs (Growth + Non-growth) are listed on Fig. 3a. and 36 TFs in supplementary table 7. In addition, they addressed that 15 control non-targeting sgRNAs and 16 sgRNAs targeting essential genes were used for Spear-ATAC on Fig. 3a. But, in supplementary table 7, the numbers are 14 for non-targeting sgRNAs and 12 for sgRNAs targeting essential genes. Authors should check and correct the number of targeting genes.

We thank the Reviewer for catching these errors and we have checked that all numbers of genes and sgRNAs listed throughout the text are correct. The correct number of genes targeted in the larger screen is 40 (36 transcription factors and 4 essential genes), and the screen included 14 non-targeting controls and 12 sgRNAs targeting essential genes. We apologize for these initial inaccuracies.

- CDC5L is not shown on Fig. 3a but in Fig. 3b-c. Authors need to add CDC5L.

We thank the Reviewer for catching this error and have now added CDC5L to Fig. 3a.

Point-by-point response
NCOMMS-20-43842-T

Page 8, Line 161-163: “Overall, we captured 32,832 nuclei representing 128 sgRNA genotypes across six Spear-ATAC samples, with on average 372 single cells being assigned to each sgRNA target with high specificity (Extended Data Fig. 7a).”

- Why GM12878 and MCF7 showed relatively poor numbers of nuclei with high sgRNA assignment specificity, comparing with K562?

The number of nuclei assigned to sgRNAs is dependent on several factors, including the number of nuclei captured from a single sample as well as the number of samples processed overall. Users can choose to target up to 10,000 nuclei in a single 10x scATAC-seq sample. We targeted and captured fewer cells for the GM12878 and MCF7 cell lines and also used fewer samples for these cell lines compared to K562 cell lines; therefore, we have fewer nuclei associated to sgRNAs. We have clarified this in the Methods (pages 16, lines 332-341):

“On average, 6000 nuclei were targeted per sample for Spear-ATAC for K562;dCas9-KRAB samples and 4000-5000 nuclei were targeted per sample for Spear-ATAC for MCF7;dCas9-KRAB and GM12878;dCas9-KRAB samples. For the pilot screen, one sample was processed for K562;dCas9-KRAB. For the time-course screen, four samples were processed for K562;dCas9-KRAB (one for each time-point). For the large screen, six identical samples were processed in parallel for K562;dCas9-KRAB and four identical samples were processed in parallel for MCF7;dCas9-KRAB and GM12878;dCas9-KRAB samples.”

We have further clarified in the text that the screen in MCF7 and GM12878 cells was a smaller scale screen than the one performed in K562s (page 9, lines 178-182):

“To further test the specificity and performance of Spear-ATAC in additional cell lines, we performed a smaller scale screen with the same K562-optimized sgRNA targets in GM12878;dCas9-KRAB lymphoblastic cells and MCF7;dCas9-KRAB breast cancer cells.”

Page 9, Line 167-170: “Similarly, for sgRNA genotypes that resulted in strong motif accessibility differences compared to sgNT-containing cells, the motifs identified were often consistent with the targeted transcription factor, as shown for GATA1, NFE2, KLF1, FOSL1, and NRF1 (Fig. 3c).”

- What kinds of the motifs identified were consistent or inconsistent with the targeted TFs? Also, authors should explain why such a difference occurred.

We thank the Reviewer for this question. We did not identify motifs that were inconsistent with the targeted TFs – however, for sgRNA genotypes that did not result in strong motif accessibility differences compared to sgNT-containing cells, we did not have enough statistical power to call motif differences. We have re-phrased this sentence and taken out the word “often” which was misleading.

Reviewer #2 (Remarks to the Author):

The authors describe a method to quantify and map the effects of perturbing transcription factor levels on chromatin accessibility in high throughput. Spear-ATAC (Single-cell perturbations with an accessibility read-out using scATAC-seq) is a droplet-based single-cell ATAC-seq protocol that enables simultaneous read-out of chromatin accessibility profiles and integrated sgRNA spacer sequences from thousands of individual cells at a time. It relies on reading out sgRNA spacer sequences directly from gDNA rather than from RNA transcripts.

Point-by-point response NCOMMS-20-43842-T

As the authors discuss, this is not the first time that a method combining CRISPR screening with chromatin accessibility profiling of single cells is described. The previously published Perturb-ATAC (Rubin et al., 2019, Cell) is limited to analyzing 96 cells per run on a microfluidic device. Spear-ATAC promises to increase the screening throughput by at least 35-fold and to decrease the experimental cost by 20-fold. Also the fact that it is based on 10x genomics technology for capturing single nuclei, makes this protocol attractive for many labs that are currently using 10x genomics for scATAC-seq analysis.

We thank the Reviewer for the positive assessment of our work.

Given that Spear-ATAC protocol may be used by labs with no strong background in developing such protocols (but rather rely on manuscripts like the one under review) it is important that the authors put some more effort in describing the protocol in more details. Fig 1a and extended Data Fig. 1a is a good start, but not very detailed. Especially more details should be given on the sgRNA plasmid. The authors provide a detailed protocol only for the targeted amplification of sgRNAs (Supplementary Note 1). It would be important that they do so for all steps of Spear-ATAC (from cloning and virus production to data analysis). The same applies to code availability. The authors promise to provide custom code used upon request, but they may want to make it available via Github etc.

We agree and thank the Reviewer for this suggestion and agree this would be a valuable addition. We have now included a more detailed Supplementary Note outlining the protocol for cloning Spear-ATAC plasmids (Supplementary Note 1), making virus and transducing cells (Supplementary Note 2), as well as the original protocol for targeted amplification of sgRNAs (which is now Supplementary Note 3). With regard to code availability, we have now made a Github for this manuscript with relevant code provided, and also generated a tutorial for applying the code to a “canned” data set (https://github.com/GreenleafLab/SpearATAC_MS_2021). We too hope these additions will facilitate the adoption of this technology by diverse labs who do not necessarily have a background in technology development.

There is some misleading regarding the number of cells profiled by Spear-ATAC. In the abstract it is mentioned that 104,592 cells (representing 414 sgRNAs) were profiled, giving the impression that this cell number comes from a single Spear-ATAC run. In a similar manner the authors comment in the manuscript that Spear-ATAC allows to profile up to 80,000 nuclei at once (line 72). However, the highest number of cells processed at once in this manuscript is 32,832. Is this correct? The authors may want to clarify this issue. What is the number of fragments sequenced per cells assigned to each sgRNA?

We thank the Reviewer for prompting us to clarify these numbers in the text. The wording of our abstract is consistent with how other methods have summarized their manuscripts; for example, the abstract of Perturb-ATAC mentions that “we applied Perturb-ATAC... in 4,300 single cells, encompassing more than 63 genotype-phenotype relationships” and the abstract of Perturb-seq mentions that “We demonstrate Perturb-seq by analyzing 200,000 cells in immune cells and cell lines.” In both of these instances, these cell counts are a summarization of many experiments, and not a single run of these methods. For our own manuscript which is worded similarly, we have tried to clarify by saying “across three experiments” and do hope that clarifies this issue (see page 2, lines 5-6).

With regards to how many cells can be profiled at once using Spear-ATAC, it is correct that we processed 32,832 cells across six Spear-ATAC samples (one 10x Controller run) when we could have processed up to 60,000 cells (10,000 cells per sample) in the same run. Instead

Point-by-point response NCOMMS-20-43842-T

of choosing to capture the maximum number of nuclei per sample recommended by 10x Genomics (10,000 – or up to 80,000 per 10x Controller run with 8 samples), we targeted 6000 nuclei per run and captured approximately the number that we targeted. There are pros and cons to targeting fewer and more nuclei, and we have now included more information in the Methods section to help users decide how many nuclei to target depending on the experiment (pages 15-16, lines 321-332):

“10x Genomics recommends that users capture between 500-10,000 nuclei from each sample. If more than 10,000 nuclei are loaded into a single sample, it increases the likelihood of clogging the microfluidic channel within the chip and is not recommended. Sometimes a user might be limited based on the number of nuclei available and might only choose to capture 500 nuclei total; other times the number of nuclei might not be limiting, but a user will still choose to target only 6000-7000 nuclei rather than the maximum number of 10,000 nuclei. The main downside to targeting more nuclei is that the multiplet rate (the number of gel bead emulsions that will be loaded with more than one nucleus) will increase – if a user targets 5000 nuclei, the multiplet rate is ~3.9%. If a user targets 10,000 nuclei, the multiplet rate is ~7.6% (according to the 10x Genomics website.”

In addition, we have included a new supplemental figure (Supplementary Fig. 5) to show readers how the cost and throughput scales according to the number of nuclei captured as well as the % of nuclei assigned to a sgRNA.

Line 97: The authors suggest that GATA3 is activated upon GATA1 kd but this is not shown at the expression level. An ATAC-seq peak in the gene body of GATA3 after GATA1 kd does not necessarily mean increase in the expression levels of the corresponding gene.

We thank the Reviewer for this question. We attempted to carry out qPCR for GATA3 following GATA1 knockdown, but it appears that GATA3 is either not expressed or expressed at very low levels in GATA1-knockdown cells. We have now taken out the two panels relating to this observation to avoid future confusion.

While this manuscript was under review, a manuscript with a similar focus was posted on bioRxiv (Liscovitch-Brauer et al) <https://www.biorxiv.org/content/10.1101/2020.11.20.390971v1>. The authors may comment on this method and how it compares to Spear-ATAC in the discussion.

We thank the Reviewer for this suggestion and agree this work deserves discussion. We have now added a section to the Discussion paragraph to highlight a key difference between these two methods (page 10 lines 194-201):

“Of note, while this manuscript was under review, a preprint of another method to capture sgRNAs from RNA transcripts alongside scATAC-seq was made available (CRISPR-sciATAC). In contrast to both Perturb-ATAC and Spear-ATAC, CRISPR-sciATAC relies on using 96 barcoded transposases to index ATAC and cDNA fragments from the same nucleus in plates. While the throughput of Spear-ATAC and CRISPR-sciATAC are similar, we anticipate that Spear-ATAC may be easier for laboratories to adopt due to its use of commercially available reagents.”

Point-by-point response
NCOMMS-20-43842-T

References

- Buenrostro, J.D., Wu, B., Litzenburger, U.M., et al. 2015. Single-cell chromatin accessibility reveals principles of regulatory variation. *Nature* 523(7561), pp. 486–490.
- Datlinger, P., Rendeiro, A.F., Schmidl, C., et al. 2017. Pooled CRISPR screening with single-cell transcriptome readout. *Nature Methods* 14(3), pp. 297–301.
- Dixit, A., Parnas, O., Li, B., et al. 2016. Perturb-Seq: Dissecting Molecular Circuits with Scalable Single-Cell RNA Profiling of Pooled Genetic Screens. *Cell* 167(7), p. 1853–1866.e17.
- Fulco, C.P., Munschauer, M., Anyoha, R., et al. 2016. Systematic mapping of functional enhancer-promoter connections with CRISPR interference. *Science* 354(6313), pp. 769–773.
- Jost, M., Santos, D.A., Saunders, R.A., et al. 2020. Titrating gene expression using libraries of systematically attenuated CRISPR guide RNAs. *Nature Biotechnology* 38(3), pp. 355–364.
- Rubin, A.J., Parker, K.R., Satpathy, A.T., et al. 2019. Coupled Single-Cell CRISPR Screening and Epigenomic Profiling Reveals Causal Gene Regulatory Networks. *Cell* 176(1–2), p. 361–376.e17.

Reviewers' Comments:

Reviewer #1:

Remarks to the Author:

the authors have adequately responded to reviewer comments. I recommend publication.

Reviewer #2:

Remarks to the Author:

The authors have comprehensively addressed my comments in the revised version of the manuscript.